# Anisotropic etching of platinum electrodes at the onset of cathodic corrosion

Thomas J.P. Hersbach[1], Alexei I. Yanson[2] & Marc T.M. Koper[1]

Cathodic corrosion is a process that etches metal electrodes under cathodic polarization. This process is presumed to occur through anionic metallic reaction intermediates, but the exact nature of these intermediates and the onset potential of their formation is unknown. Here we determine the onset potential of cathodic corrosion on platinum electrodes. Electrodes are characterized electrochemically before and after cathodic polarization in 10 M sodium hydroxide, revealing that changes in the electrode surface start at an electrode potential of −1.3 V versus the normal hydrogen electrode. The value of this onset potential rules out previous hypotheses regarding the nature of cathodic corrosion. Scanning electron microscopy shows the formation of well-defined etch pits with a specific orientation, which match the voltammetric data and indicate a remarkable anisotropy in the cathodic etching process, favouring the creation of (100) sites. Such anisotropy is hypothesized to be due to surface charge-induced adsorption of electrolyte cations.

[1] Leiden Institute of Chemistry, Leiden University, PO Box 9502, Leiden 2300 RA, The Netherlands. [2] Cosine Measurement Systems, Oosteinde 36, Warmond 2361 HE, The Netherlands. Correspondence and requests for materials should be addressed to M.T.M.K. (email: m.koper@lic.leidenuniv.nl).

Cathodic corrosion is a phenomenon in which metal electrodes undergo degradation under cathodic conditions. This process has puzzled scientists since its discovery in 1902 by Haber because of the unexpected changes that are induced on the electrode surface at negative potentials[1]. Besides leading to extensive roughening of the surface, cathodic corrosion also generates nanoparticles as a corrosion by-product at strong cathodic polarization, which can be enhanced by introducing an alternating potential. These observations have led researchers to hypothesize incorporation and subsequent leaching of electrolyte protons or alkali metals as an explanation for cathodic corrosion[1–3]. This process would weaken the structure of the metal lattice, leading to degradation of the surface and formation of nanoparticles. Another possible mechanism suggested for the formation of these particles was 'contact glow discharge', a phenomenon induced by high currents, which has the potential to rapidly degrade electrodes[4].

However, we demonstrated that cathodic corrosion even takes place if the electrolyte cations are organic instead of alkali metals, if anodic potentials are not applied, and if the measured currents are far below those required for contact glow discharge[5]. Another important observation from our previous work was that cathodic corrosion does not take place if protons are the only cations in solution. These observations rule out the aforementioned hypotheses and instead point exclusively to cathodic chemical reactions being the main reason for corrosion. On the basis of these observations, we suggested cathodic corrosion to occur via metastable metallic anions, which are stabilized by non-reducible electrolyte cations. The exact nature of these corrosion intermediates and the exact cathodic corrosion onset potential are, however, still unknown; most recent studies on cathodic corrosion have employed a practical approach towards nanoparticle synthesis[6–11], rather than a fundamental approach towards understanding cathodic corrosion. Additionally, these studies often did not employ reference electrodes and generally applied high-amplitude AC voltages, thereby impairing the ability to draw clear conclusions on the exclusive role of cathodic potentials.

In pursuit of elucidating the processes underlying cathodic corrosion, this work focuses on studying the onset of cathodic corrosion at platinum electrodes by detailed electrochemical and structural characterization. The electrodes are subjected to various constant cathodic potentials in a 10 molar sodium hydroxide solution and are subsequently characterized by cyclic voltammetry (CV). This CV analysis reveals a remarkable anisotropic etching with an onset potential only a few hundreds of millivolts negative of the onset of hydrogen evolution. These findings of anisotropic etching are supported by scanning electron microscopy (SEM), which reveals the formation of well-oriented etch pits. Our results match well with the previously observed preferred orientation of nanoparticles prepared by cathodic corrosion. This matching preferred orientation is hypothesized to be caused by the specific interaction of electrolyte cations, which are known to be crucial actors in the cathodic corrosion process[5]. Furthermore, the strategy employed in this work seems suitable for studying cathodic corrosion on other metals.

## Results

**Surface changes observed by cyclic voltammetry.** To determine the onset potential of cathodic corrosion, we followed the following protocol. Platinum work electrodes were polarized cathodically at a given constant potential for 60 s in a 10 M NaOH solution. These cathodic potentials were applied versus an internal reversible hydrogen electrode (RHE). Before and after polarization, the electrodes were characterized by CV in the hydrogen adsorption/desorption region to determine whether the electrode surface had changed. CV is a quick and versatile technique for characterizing platinum electrodes, since potential induced adsorption and desorption of hydrogen on platinum in sulfuric acid are extremely sensitive to the orientation of the atoms on the electrode surface. Therefore, different surface sites produce different peaks in the voltammogram. Specifically, (100) sites near terrace borders produce a relatively sharp peak at 0.27 V versus RHE, whereas (100) terrace sites are responsible for a broad signal between 0.3 and 0.4 V versus RHE[12]. In addition, (111) sites give a broad featureless signal between 0.06 and 0.3 V versus RHE due to hydrogen adsorption, along with another broad feature between 0.4 and 0.55 V versus RHE due to sulfate adsorption. Finally, (110) sites generate a peak at 0.13 V versus RHE. These peaks will be visible and distinguishable in the voltammogram if their corresponding surface sites are present on the electrode. Voltammograms for several studied platinum electrodes are displayed in Fig. 1.

As can be seen in Fig. 1a, treating platinum at a potential of −0.3 V versus RHE leads to voltammograms that overlap almost perfectly before and after polarization. Only a small increase in the (110) peak at 0.13 V versus RHE and a minor decrease in the (100) peak at 0.27 V versus RHE are observed after treatment, but these changes are minimal and occur consistently for all tested potentials above and including −0.3 V versus RHE.

Equally subtle, yet reproducible changes in the voltammogram occur when the electrode is treated at −0.4 V versus RHE. These changes are observable in Fig. 1b and are marked by a small decrease in the (110) peak, accompanied by a small increase in the (100) peak. Moreover, a marginally higher current is observed between 0.3 and 0.4 V versus RHE. These changes in the voltammogram imply an increase in the number of (100)-type sites.

This surface modification towards (100) sites is much more apparent after the electrode has been polarized at −0.5 V versus RHE, as visualized in of Fig. 1c. The number of (100)-type sites has increased dramatically and (110) sites have almost completely disappeared; virtually all current at 0.13 V versus RHE originates from the broad feature between 0.06 and 0.3 V versus RHE, which corresponds to atoms arranged in a (111)-type fashion. The abundance of these (111)-type sites has increased slightly, as can be derived from the clearly increased (bi)sulfate adsorption feature between 0.4 and 0.55 V versus RHE.

All changes described for −0.5 V versus RHE polarization are even further enhanced if platinum is polarized at −1.0 V versus RHE. Most notably, the (100) peak at 0.27 V versus RHE has grown strongly and a peak corresponding to wide (100)-type terraces has developed at 0.38 V versus RHE. In addition, the amount of (111)-type sites has increased, as is indicated by an increase in the broad current features corresponding to these sites. Finally, the total charge corresponding to both the cathodic and anodic CV signals has increased by a factor of 1.6. This correlates with a factor 1.6 surface area increase[13], which indicates significant roughening of the electrode surface.

**Scanning electron microscopy.** Since the CV data indicate changes in the arrangement of atoms on the surface and roughening of the electrode, one expects to observe a change in surface morphology from inspection of the surface. The surface can be imaged by, for example, in situ scanning tunnelling microscopy (STM) or by ex situ SEM. Although STM is capable of achieving atomic resolution on well-defined surfaces in electrochemical systems[14,15], obtaining such resolution during cathodic corrosion is prohibited by a variety of factors. Most notably, vigorous evolution of hydrogen during corrosion

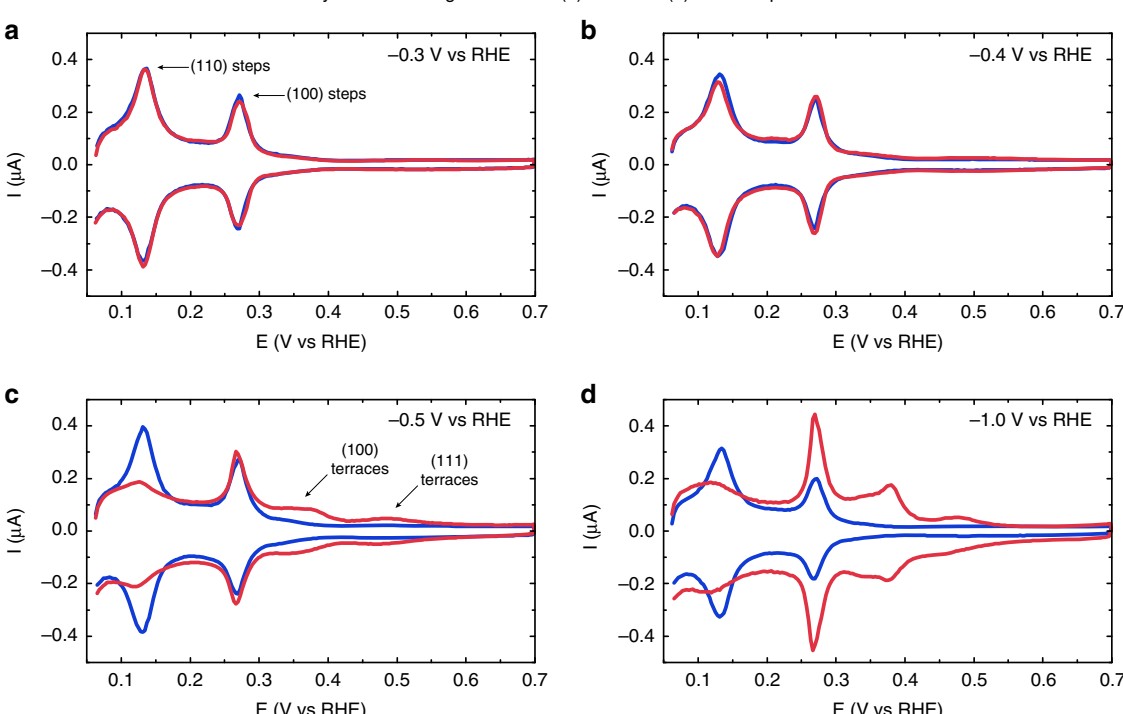

**Figure 1 | Cyclic voltammetry.** Cyclic voltammograms of platinum electrodes before (blue trace) and after (red trace) cathodic polarization in 10 M NaOH at − 0.3 V versus RHE (**a**), − 0.4 V versus RHE (**b**), − 0.5 V versus RHE (**c**) and − 1.0 V versus RHE (**d**). Voltammograms were recorded in 0.5 M $H_2SO_4$, at a scan rate of 50 mV s$^{-1}$.

prevents imaging of the electrode[16]. In addition, a wide range of challenges is posed by the switching between 0.5 M $H_2SO_4$ and 10 M NaOH electrolyte that would be required to characterize the electrode before and after cathodic treatment. To our best knowledge, resolving these challenges is currently beyond the state-of-the-art.

Therefore, *ex situ* SEM is a more easily accessible technique, which provides valuable information in addition to CV characterization. Typical SEM images are shown in Fig. 2, which displays micrographs of electrodes treated at various potentials between − 0.2 and − 0.8 V versus RHE. It is vital to realize that these micrographs have a lower resolution than STM and cannot visualize the smallest conceivable electrode roughness. Though nanoscale corrugation is present on even the most well-defined single crystals[17], observing such height differences is beyond the practical resolution of the employed microscope. Still, the images in Fig. 2 show excellent consistency with the conclusions from the CV measurements, as will be discussed in the next paragraphs.

The SEM image in Fig. 2a shows the platinum surface polarized at − 0.2 V versus RHE, displaying only barely visible degrees of roughness in the bottom right quadrant. Apart from the three intersecting crystal grain boundaries, the depicted area can therefore be considered to be mostly flat from an SEM point of view. This matches with the observations made in CV, in which electrodes look identical when freshly annealed and polarized at or above − 0.3 V versus RHE. Similarly, no observable roughening of the electrode can be seen after polarizing it at − 0.4 V versus RHE (Fig. 2b). Even when decreasing the potential to − 0.5 V versus RHE (Fig. 2c), where voltammetry detects a clear change in surface morphology, only a barely distinguishable roughening can be observed.

Significantly more roughening is observed after polarization at potentials of − 0.6 V versus RHE or lower; the micrograph in Fig. 2d clearly depicts increased corrugation, both at the crystal grains and their boundaries. Such corrugation is present on most

parts of the electrode, which agrees well with an electrochemically determined roughness factor of 1.1 for the electrode in Fig. 2d,e and 1.2 for the electrode in Fig. 2f.

Interestingly, this general increase in corrugation is accompanied by the formation of well-defined etch pits on parts of the electrode. One type of etch pit is triangular, as shown in Fig. 2e. Three of these pits have been outlined in yellow. Notably, all pits are oriented identically if they are on the same crystal grain.

The second type of pits is depicted in Fig. 2f and exclusively possesses 90° angles. Because of their shape, the pits can be seen as quasi-rectangular; similar pits have been observed previously in AC corrosion[18]. These pits are also oriented identically when they are formed on the same grain, as can readily be seen by comparing the three highlighted pits. The pits and corrugation described here are likely the only cause of the electrochemically observed surface area increase, since no cathodically formed nanoparticles were observed.

## Discussion

The CV data presented in Fig. 1 show that the electrode surface structure remains unchanged when platinum wires are polarized at potentials of − 0.3 V versus RHE or higher in a 10 M NaOH solution. However, the surface structure is modified when the electrode is polarized at potentials of − 0.4 V versus RHE and below. We can therefore conclude that the onset potential of cathodic corrosion of platinum lies between − 0.3 and − 0.4 V versus RHE. Since determining the exact onset potential requires an even greater accuracy than the current experimental setup provides, we suggest a tentative cathodic corrosion onset potential of − 0.4 V versus RHE for platinum in a 10 M NaOH solution. This corresponds to approximately − 1.3 V versus the normal hydrogen electrode (NHE).

This experimentally determined onset potential presumably has a thermodynamic character because it is determined by the

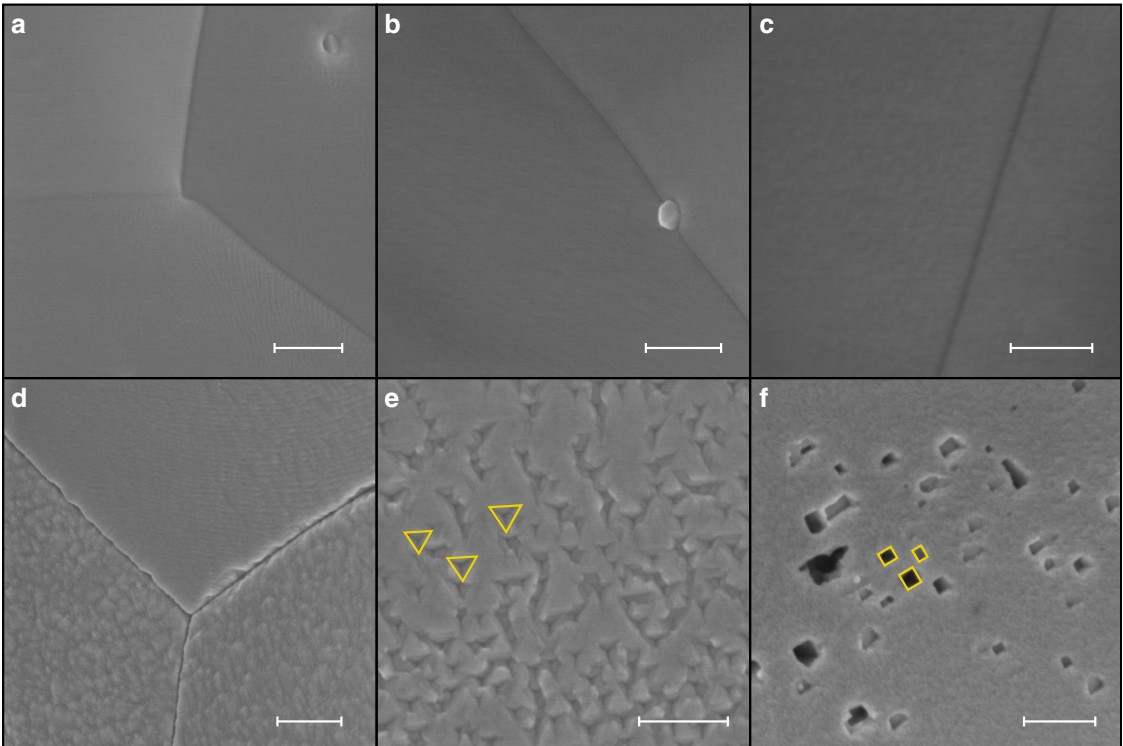

**Figure 2 | Electron microscopy.** Scanning electron micrographs of platinum electrodes treated at − 0.2 V versus RHE (**a**), − 0.4 V versus RHE (**b**); − 0.5 V versus RHE (**c**); − 0.6 V versus RHE (**d**,**e**) and − 0.8 V versus RHE (**f**). In **e**,**f**, three etch pits have been outlined in yellow to illustrate shape and orientation similarities. Scale bars, are 300 nm (**a**,**b**,**d**–**f**). Scale bar, 100 nm (**c**).

stability of the elusive metastable corrosion intermediate. However, this onset potential is not a standard equilibrium potential, such as those listed in the electrochemical series[19]; in order to define an equilibrium potential, one would require accurate knowledge of the concentration and nature of the involved reactants. Since this knowledge is currently unavailable due to the elusive nature of the cathodic corrosion reaction intermediates, the reported onset potential is simply the least negative potential at which cathodic corrosion can be detected. We can therefore not exclude that longer corrosion times would slightly shift the determined onset potential to less negative values, leading to the conclusion that this potential is also partly kinetic in nature.

Still, the value of the onset potential can be compared with tabulated equilibrium potentials. For example, the onset potential lies only 0.4 V below the thermodynamic onset of hydrogen evolution in alkaline media, which is surprisingly mild. This potential of − 1.3 V versus NHE definitively rules out the incorporation of sodium ions as the reason for cathodic corrosion; the $Na^+/Na$ couple has a standard equilibrium potential of − 2.71 V versus NHE[19].

The CV data also indicate that cathodic corrosion is accompanied by the preferential formation of (100)-type sites. This implies that etching by cathodic corrosion is highly anisotropic. These observations are confirmed by the shape and orientation of the etch pits created by cathodic corrosion, displayed in the micrographs in Fig. 2e,f. These shapes and orientations can be rationalized by the simple models shown in Fig. 3, which will be used to illustrate how the SEM data support the (100) etching preference observed in electrochemistry.

Figure 3a displays a (111)-type surface with a (110)-type step. Etching a hole in this surface that exposes (100) sites requires the hole to be triangular. This model etch pit shape matches the shape of the etch pits in Fig. 2e. Moreover, Fig. 3 indicates that all

etch pits should have the same orientation, as dictated by the orientation of the underlying crystal grain. This is indeed apparent from the identically oriented outlines in Fig. 2e. Finally, the model in Fig. 3 matches the electrochemically observed decrease in (110) sites: if an etch pit is created in a surface section with a (110)-step, part of this step will be removed. Thus, the model in Fig. 3a is able to unify the electrochemical and microscopic observations.

A similar analysis can be performed on (100)-type surfaces, displayed in Fig. 3b. When exposing (100) sites in this surface, one is required to create holes with a rectangular shape of which the orientation is again dictated by the crystal grain. Furthermore, all angles within the hole must be 90°. These identically aligned rectangles match well with those in Fig. 2f, as is emphasized by the orientation of the highlighted rectangles. Any deviation from the model rectangular shape has to add new rectangles to the existing pit in order to exclusively create (100) sites, thus creating quasi-rectangles such as the bottom left pit in Fig. 2f. In addition, any (110) sites will be removed from the surface in which an etch pit is grown. Thus, the rectangular model pits are also in good agreement with both the electrochemical and microscopic data.

One should bear in mind that the presented model pit shapes are only meant to illustrate how pit shape and voltammetry are consistent, and that the actual etching process through which the pits are generated most likely occurs through an interplay of complex mechanisms, much alike surface growth by atom deposition[20]. For example, one could imagine hole initiation by corrosion of a (110)-step site, from which the hole grows over the surface. The complexity of the corrosion kinetics is explicitly suggested by the occurrence of corrugation that does not resemble the model pit shapes presented in Fig. 3, such as the type depicted in Fig. 2d. This type of roughness on grains and grain boundaries is abundant on the electrode, as is the occurrence of complex etching grooves. These features are to be

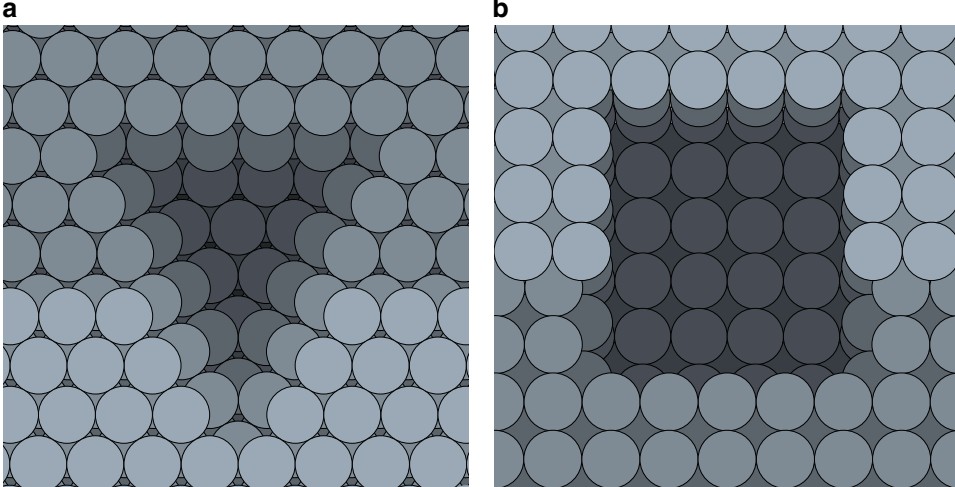

**Figure 3 | Model etch pits.** (**a**) A triangular etch pit with (100)-type sides in a (111)-type surface with a (110)-type step and (**b**) a rectangular etch pit with (100)-type sides in a (100)-type surface with a (111)-type step.

expected, since metallic surfaces possess elaborate reconstructions that are typically invisible to SEM, such as step bunching. Such reconstructions are much more complex than the idealized (111)- and (100)-type surfaces employed in our model. Explaining corrugation on these non-ideal areas of the electrode will thus require more complex kinetic models, which will be the focus of future studies of cathodic corrosion. Nonetheless, the presented model etch pits are able to unify the electrochemical and SEM data and emphasize the strong (100) etching preference of platinum.

This preference for (100) sites is remarkable, since (100)-type surfaces typically have higher surface energies than (111)-type surfaces and even (100) steps have a higher free energy than (111) steps on a (111) surface[21,22]. In addition, the preference for (100) sites is present for both the observed etch pits and the orientation of nanoparticles created by cathodic corrosion[7,8]. Any consistent explanation for this preference will therefore have to address both the anisotropy in cathodic etching and the preferential nanoparticle orientation. The preferential selection of a certain facet can be surface charge driven (global) or adsorbate driven (local).

A global explanation involving surface charge-induced reconstruction is less likely[23], since such a hypothesis would only explain the etching anisotropy. This hypothesis does not take into account the anisotropic particle growth, which necessarily occurs through coalescence of uncharged intermediates. Any anionic intermediates will have to be oxidized before coalescence, since coulombic repulsion should prevent charged particles from colliding. Similarly, global hypotheses based on the potential dependence of the free energy of different step types will only explain the etching anisotropy[24], because nanoparticle growth is thought to occur in solution and should be largely unaffected by the electrochemical potential of the electrode.

Since global explanations typically only seem to explain the etching anisotropy, a local hypothesis involving stabilization by adsorbing electrolyte cations seems more appropriate[25]. These positively charged ions are expected to interact strongly with the electrode surface at the negative potentials and surface charges relevant for cathodic corrosion. Therefore, these cations are likely able to restructure the platinum surface; it is well known that adsorbates can affect the structure of metallic surfaces by lowering the surface free energy of facets that would not be preferred under adsorbate-free conditions[26–28]. Furthermore, it is known that these cations are essential to cathodic corrosion, since they are required to stabilize the anionic

corrosion intermediates[29]. This importance of cations parallels the role of adsorbates in traditional nanoparticle synthesis, in which cationic, anionic and molecular adsorbates are able to both restructure existing nanoparticles and control particle shape during formation by preferentially adsorbing to specific crystal facets[30–32].

The above mechanism would imply a strong dependence of the anisotropy in cathodic corrosion on both the concentration and nature of the electrolyte cations, which has indeed been observed in previous studies: the (100) nanoparticle orientation preference decreases with decreasing cation concentration[18,25] and is less pronounced in potassium hydroxide than in sodium hydroxide[8]. On the basis of these experimental observations and the general tendency of adsorbates to restructure nanoparticles and bulk electrodes, we hypothesize that the specific interaction of cations is the most likely cause of both the preferential orientation of cathodically prepared nanoparticles and the anisotropic etching observed in the current work.

Finally, it is interesting to compare the surface modifications observed in cathodic corrosion to other electrochemical surface modifications. For example, Díaz et al. observed striking changes in the voltammetric profile of platinum after cathodic polarization in sulfuric acid solutions, which they attributed to the formation of 'superactive' platinum states[33,34]. Although experiments in ultraclean sulfuric acid demonstrated that these observations are not caused by cathodic corrosion[5], these changes further illustrate the pronounced modifications that can occur at metallic electrodes after cathodic polarization.

The significance of the changes caused by cathodic corrosion is emphasized by comparing them to modifications caused by repeatedly cycling at predominantly anodic potentials[35–37]. These latter cycling procedures can modify electrode surfaces by repeatedly oxidizing and reducing platinum in experiments that typically last 5 –10 min. These oxidation/reduction cycles are a strict requirement for modification in these experiments, because platinum is quite stable under constant anodic polarization due to protection by a thin oxide layer[38]. By contrast, cathodic corrosion is able to induce similar or even more dramatic changes by polarizing the electrode just 0.5 V below the thermodynamic onset of hydrogen evolution for only a minute. This leads to the surprising and counterintuitive conclusion that cathodic corrosion can, in some electrolytes, be much more detrimental to platinum electrodes than anodic corrosion, suggesting that the concept of cathodic protection is relative.

Summarizing, we have determined the onset potential of cathodic corrosion of platinum in 10 M NaOH at −1.3 V versus NHE. In addition, cathodic corrosion was shown to involve highly anisotropic etching, favouring the creation of (100) terraces and steps and the removal of (110) sites. Accordingly, SEM revealed well-oriented etch pits, confirming the anisotropic etching that was determined electrochemically. Our current understanding of the phenomenon suggests that this anisotropy is likely induced by strong interaction of electrolyte cations with the highly negative surface charge of the electrode. The fact that cathodic corrosion starts only 0.4 V below the thermodynamic onset of hydrogen evolution in alkaline media also leads to the surprising conclusion that cathodic protection is a relative concept, and that cathodic corrosion can be more detrimental to platinum surfaces (and noble metals in general) than anodic corrosion.

## Methods

**Electrochemical characterization and cathodic polarization.** Electrochemical experiments were performed with an Autolab PGSTAT12, which is equipped with a linear scan generator. All water used for rinsing and preparing electrolyte solutions was demineralized and ultrafiltered by a Millipore MilliQ system (resistivity > 18.2 MΩ cm, TOC < 5 p.p.b.). All electrolyte solutions were deoxygenated before experiments by purging with argon (Linde, 6.0 purity). Argon was kept flowing over the solution during experiments.

Platinum wires (Mateck, 99.99%; ø = 0.1 mm) were used as working electrodes. Before experiments, the electrode was carefully rinsed with water and subsequently flame-annealed for 60 s and cooled down in air. Next, it was inserted into a standard three-electrode cell containing 0.5 M $H_2SO_4$ (Merck, Ultrapur), using a platinum wire as the counter electrode and a RHE as a reference electrode. The immersion depth of the working electrode was carefully controlled, using a micrometre screw. After immersion, the electrode was characterized using CV.

After CV characterization, the working electrode was rinsed and transferred to a homemade fluorinated ethylene propylene cell containing a 10 M NaOH (Fluka, Traceselect) solution, which was outfitted with a titanium counter electrode and a HydroFlex RHE (Gaskatel). Following electrode immersion, a constant cathodic potential was applied for 60 s, after which the electrode was removed under potential control. Next, the electrode was rinsed, transferred back to the $H_2SO_4$ cell and characterized again using CV. After electrochemical characterization, electrodes were removed from the cell, rinsed and stored for later examination using SEM.

**Scanning electron microscopy.** Micrographs were obtained on a FEI NOVA NanoSEM 200 SEM, using an acceleration voltage of 5 kV and a beam current of 0.9 nA. Storage time before SEM imaging did not affect the electrode, as similar images could be obtained after both several days and several months of storage.

**Data availability.** The data supporting the findings of this study are available from the corresponding author upon request.

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

## Author contributions

T.J.P.H., A.I.Y. and M.T.M.K. conceived and designed the experiments. T.J.P.H. performed all the experiments. All authors contributed to the interpretation of the data. T.J.P.H. and M.T.M.K. co-wrote the paper.

## Additional information

**Competing financial interests:** The authors declare no competing financial interests.

