## [Peer Review File · Nature Communications]

Reviewers' comments:

Reviewer #1 (Remarks to the Author):

In this paper, Hersbach et al. reported an impressive study of the anisotropic etching of Pt electrodes at the onset of cathodic corrosion. This study addresses a puzzle of many years on the nature of cathode corrosion. It provides depth understanding of the issue at a level that previously has not been achieved. Many details reported in the paper, including the onset of potential and the anisotropic cathodic etching, are highly valuable information that would benefit many readers in the field. I support publication of this paper in Nature Comm., after the follow questions/comments are addressed.

1. Is the onsite potential affected by the voltage scanning rate? Or, is it a thermodynamically controlled process or kinetics dominant?

2. It is unfortunate that the resolution of the SEM paper is very limited. The etch "pits" look like not regular.. It is not clear how the conclusion of "(100) is favorable" was drawn.

3. This paper suggested a very interesting reaction pathway that the anisotropic etching might be due to surface charge-induced adsorption of electrolyte cations. This is a significant enough question that the authors are encouraged to provide experimental evidence

Reviewer #2 (Remarks to the Author):

This manuscript is an extension of the authors' previous work and reports on the effect of cathodic polarisation of polycrystalline Pt wires in 10 M NaOH where it was found that Pt is selectively etched at surprisingly mild applied potentials of 0.4 V below the thermodynamic onset of hydrogen evolution in alkaline media. This etching was found to be anisotropic and resulted in the formation of (100)-type sites and removal of (110) sites. The signature voltammetric responses associated with these sites was correlated with SEM imaging. Overall the manuscript reports on a very interesting phenomenon which has only received fleeting attention in the literature. There are some points however that should be considered

There should be some discussion on the work by Diaz et al. namely J. Coll. Interfac. Sci. 313, 2007, 232 and Int. J. Hyd. Energy 34, 2009, 3519 where they reported the cathodic polarisation of Pt in acidic solution and observed voltammetric peaks at similar potentials to those reported here. Although the interpretation of these peaks by Diaz is significantly different, I feel it should still be discussed.

In the experimental section it appears the electrochemically polarised electrodes were characterised almost immediately in acidic solution and then stored for SEM imaging. Have the authors considered what may happen to the surface with time between CV characterisation and imaging? Do you get the same CV profile after SEM imaging as seen in Figure 1?

Although it is clear from the SEM images that etching occurs, more information would be achieved with STM imaging and may give a better indication of the etching process. A roughness value would also be informative.

Have the authors checked the 10 M NaOH solution post polarisation for the presence of any Pt ions by ICPMS? Are all the etched Pt species redeposited during the experiment?

This is an excellent piece of work and offers significant insights into quite an interesting phenomenon.

Anthony O'Mullane

Reviewer #3 (Remarks to the Author):

The authors have found a definite correlation between the applied potential and the onset of cathodic corrosion for Pt in strongly alkaline media. Their voltammetric/morphological data data has also provided evidence for preferential etching along specific crystal directions. Although mildly interesting, the interpretation of the phenomenon observed lacked sufficient depth to warrant publication in this high impact journal. Particularly shallow is the use of ex situ scanning electron microscopy instead of in situ scanning tunneling microscopy to gain insight into more details aspects of the corrosion process. This reviewer would urge the authors to devote time and effort to get to the bottom of the underlying physicochemical factors responsible for this century old observation as opposed to publishing their findings in a piecemeal fashion.

Dear Dr. West,

We sincerely appreciate the useful and thought-provoking comments we received from the reviewers of our manuscript NCOMMS-16-06717, called "Anisotropic etching of platinum electrodes at the onset of cathodic corrosion". We have edited the manuscript to address their helpful suggestions. Below, we will outline the changes we made and reply to the reviewers' comments in more detail. All changes in the revised manuscript have been highlighted with a yellow background.

Response to Reviewer #1:

According to Reviewer #1, "In this paper, Hersbach et al. reported an impressive study of the anisotropic etching of Pt electrodes at the onset of cathodic corrosion. This study addresses a puzzle of many years on the nature of cathode corrosion. It provides depth understanding of the issue at a level that previously has not been achieved. Many details reported in the paper, including the onset of potential and the anisotropic cathodic etching, are highly valuable information that would benefit many readers in the field. I support publication of this paper in Nature Comm., after the follow questions/comments are addressed."

1. "Is the onsite potential affected by the voltage scanning rate ? Or, is it a thermodynamically controlled process or kinetics dominant?"

Response: The reviewer raises an interesting question that was indeed not addressed in the original manuscript. It is important to note that the onset potential is not affected by the scan rate, because the working electrodes were kept under constant polarization. A scan rate does thus not apply to this polarization. The scan rate in the voltammograms before and after polarization should not affect the ability to determine the onset potential, since the adsorption of hydrogen on platinum is a fast and reversible process; as long as the scan rate is the same in both voltammetric scans, small changes in the electrode surface can be detected. In our understanding, the measured onset potential would be an equilibrium potential if we had detailed knowledge of the concentration and nature of the reaction product, which is not available. The found onset potential is the least negative value at which changes in the electrode surface can be detected. On the one hand, the onset is thermodynamic in the sense that it is determined by the stability of (cation-stabilized) metastable negative intermediates of which the identity is unknown. On the other hand, it is kinetic in the sense that it could well be that we would determine a slightly less negative onset potential if we polarized the working electrode for a longer time. We have edited the 'discussion' section of the manuscript in order to reflect this consideration.

2. "It is unfortunate that the resolution of the SEM paper is very limited. The etch "pits" look like not regular.. It is not clear how the conclusion of " (100) is favorable" was drawn."

Response: The reviewer understandably points out that it can be difficult to assess the shape and orientation of the etch pits. This is especially true for the triangular pits. We have therefore highlighted three representative examples of each pit in the micrographs, so their important features should be more visible. Furthermore, we have edited the 'discussion' section of the manuscript in order to stress the fact that the (100) etching preference is demonstrated by voltammetry, but confirmed by the shape and orientation of the observed etch pits.

3. "This paper suggested a very interesting reaction pathway that the anisotropic etching might be due to surface charge-induced adsorption of electrolyte cations. This is a significant enough question that the authors are encouraged to provide experimental evidence"

Response: We thank the reviewer for indicating that the text does not emphasize the experimental observations supporting the role of cations in cathodic corrosion. Though this hypothesis is indeed partly founded on the theoretical considerations on the possible validity

of global and local explanations, there are experiments that suggest its validity. Specifically, previous work (Duca *et al.*, Topics in Catalysis 517, **2014**, 255) demonstrated that the (100) orientation preference of cathodically produced nanoparticles is less prominent in potassium hydroxide than in sodium hydroxide. Furthermore, other work (Yanson *et al.*, Electrochimica Acta 112, **2013**, 913; Yanson & Yanson, Low Temperature Physics 39, **2013**, 312) demonstrates that the (100) surface termination is more pronounced at higher cation concentrations. These experiments and our theoretical considerations lead us to believe that cation adsorption is the most likely cause of the (100) preference of cathodic corrosion on platinum. This hypothesis shall have to be confirmed in a follow-up study which rigorously compares the corrosion preference across a range of cations and compares its findings with density functional theory calculations.

We have modified the 'discussion' section of the manuscript in order to emphasize not only our reasoning, but also the experimental indications for the validity of this hypothesis.

Response to Reviewer #2:

According to Reviewer #2: "This manuscript is an extension of the authors' previous work and reports on the effect of cathodic polarisation of polycrystalline Pt wires in 10 M NaOH where it was found that Pt is selectively etched at surprisingly mild applied potentials of 0.4 V below the thermodynamic onset of hydrogen evolution in alkaline media. This etching was found to be anisotropic and resulted in the formation of (100)-type sites and removal of (110) sites. The signature voltammetric responses associated with these sites was correlated with SEM imaging. Overall the manuscript reports on a very interesting phenomenon which has only received fleeting attention in the literature. There are some points however that should be considered (...)

This is an excellent piece of work and offers significant insights into quite an interesting phenomenon."

1. "There should be some discussion on the work by Diaz et al. namely J. Coll. Interfac. Sci. 313, 2007, 232 and Int. J. Hyd. Energy 34, 2009, 3519 where they reported the cathodic polarisation of Pt in acidic solution and observed voltammetric peaks at similar potentials to those reported here. Although the interpretation of these peaks by Diaz is significantly different, I feel it should still be discussed."

Response: We thank the reviewer for suggesting this additional literature to cite, especially since it further underlines the importance of processes that occur negative to the onset potential of hydrogen evolution. We cite and briefly discuss these articles in our revised manuscript to emphasize their importance.

2. "In the experimental section it appears the electrochemically polarised electrodes were characterised almost immediately in acidic solution and then stored for SEM imaging. Have the authors considered what may happen to the surface with time between CV characterisation and imaging? Do you get the same CV profile after SEM imaging as seen in Figure 1?"

Response: The reviewer points out an interesting question relating to the stability of the changes induced to platinum electrodes. CV characterization after SEM imaging was unfortunately not possible because the samples were mounted on the SEM stage with carbon tape; we noticed that the SEM resolution at high magnifications was significantly improved upon fixating not only the untreated part, but also the cathodically treated section of the electrode. This leads to carbon contamination by residual tape on parts of the electrode, which would prevent us from acquiring clean voltammograms. Additionally, it is likely that organic contaminations from the air during storage and from outgassing of the carbon tape during SEM imaging will selectively block the electrochemical signal of the (100) sites. This would lead to a perceived decrease in (100) sites which does not correspond to a 'real' physical change in the electrode surface. We can, however, comment on the stability of

the cathodic electrode modifications based on SEM. No difference could be observed in SEM between samples that were stored for one day, several days or several months. In all cases, all samples with the same treatment potential produced similar micrographs. Since the reproducibility of these micrographs did not depend on storage time, we argue that storage time is not an issue and that the cathodically induced changes are stable. The independence of the SEM images on storage time prior to imaging has been added to the 'methods' section of the manuscript.

3. "Although it is clear from the SEM images that etching occurs, more information would be achieved with STM imaging and may give a better indication of the etching process. A roughness value would also be informative."

Response: We agree with the reviewer that STM imaging may be able to provide more information on the process. However, it is important to notice that the employed electrodes are wires with a very small radius of curvature instead of the flat electrodes that are typically used in STM. This high curvature impairs the resolution of the STM, which prevents us from identifying important features like step edges. In addition, *in-situ* STM of electrodes under cathodic polarization (with hydrogen evolution) will be an extremely challenging experiment. If the aim is to get a better indication of the etching process by determining the roughness of the electrode, it is possible to determine this roughness for the entire electrode by calculating the increase of the hydrogen desorption charge. This analysis indicates that the roughness of the surface has barely changed after polarization at -0.5 V vs. RHE, but increased markedly at lower treatment potentials. These roughness values have been added in the 'results' section of the text.

4. "Have the authors checked the 10 M NaOH solution post polarisation for the presence of any Pt ions by ICPMS? Are all the etched Pt species redeposited during the experiment?"

Response: The reviewer raises an interesting question regarding the possibility of dispersion of platinum into the solution. Though the dispersion of platinum nanoparticles has indeed been observed in previous studies, those studies employed much more extreme AC potentials. In our opinion, it is unlikely that nanoparticles detached into the electrolyte in the present study, since no nanoparticles were visible on the electrode; if particles were formed and detached into the solution, previous work suggests that at least some of them should have stayed on the electrode. This indicates that the etched platinum species are indeed redeposited during the experiment, which is in line with the fact that we are truly working at the onset of cathodic corrosion. Thus, the platinum concentration in the solution will likely not be above the detection limit of the ICPMS, especially because the solution will have to be neutralized (and thus diluted) before ICPMS analysis. Additionally, it is important to note that the formation of platinum cations can be excluded, because electrodes were exclusively polarized cathodically and our counter electrode was composed of titanium. Since the question regarding nanoparticle detachment is understandable in the light of previous work, we have added a sentence in the 'results' section which points out that no nanoparticles were visible.

Response to Reviewer #3:

According to Reviewer #3: "The authors have found a definite correlation between the applied potential and the onset of cathodic corrosion for Pt in a strongly alkaline media. Their voltammetric/morphological data data has also provided evidence for preferential etching along specific crystal directions. Although mildly interesting, the interpretation of the phenomenon observed lacked sufficient depth to warrant publication in this high impact journal. Particularly shallow is the use of ex situ scanning electron microscopy instead of in situ scanning tunneling

microscopy to gain insight into more details aspects of the corrosion process. This reviewer would urge the authors to devote time and effort to get to the bottom of the underlying physicochemical factors responsible for this century old observation as opposed to publishing their findings in a piecemeal fashion.”

Response: Though we agree with the reviewer that *in-situ* scanning tunneling microscopy measurements could provide a wealth of information on cathodic corrosion, setting up such *in-situ* STM experiments during cathodic corrosion is not trivial.

First of all, one would have to design an electrochemical (EC) STM or modify an existing EC-STM setup in order to resist the strongly alkaline electrolyte employed in this study. Provided that this is possible, one should then still be able to maintain a high degree of cleanliness in the setup: any trace contamination from, for example, 3d-metal impurities in glass will deposit on the electrode at the cathodic conditions and interfere with experiments. Similarly, there is the possibility of unremoved organic contaminations or organic contaminations arising from degradation of the tip coating material. These contaminations will decompose on the electrode and interfere with the experiment or adsorb to the electrode, thus severely poisoning the platinum or affecting cathodic corrosion.

Additionally, the vigorous hydrogen evolution that occurs during cathodic corrosion will prohibit EC-STM imaging, as has been mentioned in other research as well (Kim *et al.*, Langmuir 30, **2014**, 15053). The formed hydrogen will be oxidized electrochemically on the STM tip, thereby generating currents that are much larger than the tunneling current. These currents will interfere with the STM feedback mechanism, causing the tip to move away from the surface. Thus, ‘true’ *in-situ* experiments will be impossible until this issue is resolved.

One could, of course, consider doing quasi *in-situ* measurements in which the electrode is characterized in solution after polarization. Though this would not alleviate the contamination issues raised above, it would allow for characterization of the sample in the corrosion electrolyte. However, given the stability of the cathodically induced surface modifications, it is highly unlikely that one will observe different results than those presented in our manuscript.

In addition, using a quasi *in-situ* EC-STM setup will sacrifice many of the benefits of our current setup. For example, electrochemical characterization of the electrode before and after cathodic corrosion will be complicated; this would require continuous switching between (highly acidic) 0.5 M H₂SO₄ for accurate characterization and (highly alkaline) 10 M NaOH for corrosion, which would be both a safety hazard (because of the chance of both liquids mixing) and an additional opportunity for contamination. Furthermore, the scanning window is limited by the maximum range of the STM piezo element (typically less than 10 by 10 μm), while our SEM is able to quickly characterize an entire side of the (100μm wide and 1000μm long) electrode. Doing EC-STM experiments would thus provide only the possible benefit of increased resolution, while introducing a plethora of complications and drawbacks that are not present in our current setup.

The current setup allows for performing clean experiments and the characterization of many (duplicate) samples, thereby strongly improving the reliability of our setup. We therefore think that our present setup is the optimal one available, according to the current state-of-the-art. However, the reviewer suggests a valuable follow-up study that will undoubtedly generate further insight into cathodic corrosion once the abovementioned issues are solved by the scientific community.

Of course, we understand the reasoning behind the comment of Reviewer #3. We have therefore expanded the ‘results’ section of the manuscript with a justification of the use of SEM, such that it contains not only the drawbacks of SEM that were mentioned in the original manuscript, but also the provided benefits. We therefore think that the choice for the current working setup is now validated more explicitly in the manuscript.

We thank all three reviewers for taking the time to carefully study our manuscripts and provide important feedback.

Sincerely,

Marc Koper

REVIEWERS' COMMENTS:

Reviewer #2 (Remarks to the Author):

The authors have satisfactorily addressed all concerns raised by the reviewers and I recommend publication.